# Biomarkers for Monitoring Renal Damage Due to Fabry Disease in Patients Treated with Migalastat: A Review for Nephrologists

**DOI:** 10.3390/genes13101751

**Published:** 2022-09-28

**Authors:** Sebastián Jaurretche, Hernan Conde, Ana Gonzalez Schain, Franco Ruiz, Maria Victoria Sgro, Graciela Venera

**Affiliations:** 1Renal and Pancreas Transplant Department, Sanatorio Parque de Rosario, Rosario S2000, Argentina; 2Biophysics and Human Physiology, School of Medicine, Instituto Universitario Italiano de Rosario, Santa Fe S2000CTT, Argentina; 3Research Department, School of Medicine, Instituto Universitario Italiano de Rosario, Santa Fe S2000CTT, Argentina

**Keywords:** Fabry disease, renal disease, biomarker, migalastat

## Abstract

Nephropathy is a major Fabry disease complication. Kidney biopsies reveal glomerulosclerosis even in pediatric patients. The main manifestations of Fabry nephropathy include reduced glomerular filtration rate and proteinuria. In 2016, an oral pharmacological Chaperone was approved to treat Fabry patients with “amenable” mutations. Because (i) Fabry disease is a rare disorder that frequently causes kidney damage, and (ii) a new therapeutic is currently available, it is necessary to review wich biomarkers are useful for nephropathy follow-up among Fabry “amenable” patients receiving migalastat. The literature search was conducted in MEDLINE, EMBASE, SCOPUS, Cochrane, and Google academic. Prospective studies in which renal biomarkers were the dependent variable or criterion, with at least 6 months of follow-up, were included. Finally, we recorded relevant information in an ad hoc database and summarized the main results. To date, the main useful biomarker for nephropathy monitoring among Fabry “amenable” patients receiving migalastat is glomerular filtration rate estimated by equations that include serum creatinine.

## 1. Introduction

Fabry disease (FD) is an X-linked lysosomal storage disorder caused by defects in the glycosphingolipid metabolic pathway due to deficient or absent lysosomal α-galactosidase-A (α-GalA) enzyme activity [1]. This results in globotriaosylceramide (Gb3) pathological accumulation within lysosomes in a wide variety of cells, including endothelial, renal, cardiac, and neuronal cells [1]. Gb3 accumulation occurs in all of the renal cells (podocytes, mesangial, tubular, and interstitial cells, as well as vascular endothelial and smooth muscle cells) [2].

The “classic” FD phenotype is expressed in males with absent or severely reduced (<1–3% of mean normal) α-GalA activity, marked Gb3 accumulation in vascular endothelial cells, cardiomyocytes, smooth muscle cells, and podocytes [1,3]. Early symptoms, including neuropathic pain in the extremities (chronic paresthesia and severe episodic crises), gastrointestinal discomfort, hypohidrosis, angiokeratomas, and cornea verticillata typically emerge during childhood or early adulthood [1,3]. Among “classic” FD patients, childhood or adolescent onset of symptoms is followed by progressive multi-organ failure and shortening of life expectancy due to premature death [1,3].

Males with “late-onset” FD phenotypes present varying levels of residual α-GalA activity (>3% of mean normal) and late clinical manifestations usually limited to a single organ [1,3].

In heterozygous female FD patients, the spectrum of disease severity is heterogeneous and it is, in part, dependent on the GLA mutation and the X chromosome inactivation (Lyonization) profile [1,3].

Nephropathy is a major complication of FD [4]. Kidney biopsies reveal Gb3 accumulation in tubular epithelial cells and in glomerular and endothelial cells, with focal and global glomerulosclerosis as early as in the second decade of life and even in pediatric patients [4,5,6,7].

Main manifestations of Fabry nephropathy include reduced glomerular filtration rate (eGFR) and proteinuria [1,2,4,8]; “classic” affected males typically progress to end-stage renal disease (ESRD) by the fourth decade of life [9,10]. Among women with “classic” phenotype, kidney involvement occurs at an older age. In “late-onset” patients of both genders, kidney disease can be the only manifestation, or it may be absent because some “late-onset” FD phenotypes are exclusively cardiac [1].

Some FD patients have missense mutations with a reduction in overall α-GalA enzymatic activity due to reduced stability of the mutated protein, caused by protein misfolding and premature degradation, but the α-GalA catalytic activity remains conserved [11,12]. Figure 1 shows an algorithm of which patients are eligible for migalastat treatment. To correct this misfolding and to prevent premature degradation, migalastat (1-deoxygalactonojirimicin; DGJ), a small iminosugar molecule was developed (Figure 2).

Migalastat was initially identified as a competitive inhibitor of the α-GalA enzyme. However, at sub-inhibitory concentrations, a binding to the enzyme’s catalytic center can facilitate proper protein folding in the endoplasmic reticulum and appropriate trafficking to the lysosome. This resulted in an increase of enzymatic activity in Fabry patients (Figure 2).

This concept of how a small molecule can help a protein to fold correctly, allowing it to enter physiological processing pathways properly, was enunciated as a “pharmacological chaperone” [13,14]. 

Enzyme replacement therapy (ERT) is presently available in the form of agalsidase alfa (given at 0.2 mg/kg body weight every other week by intravenous Infusión) and agalsidase β (administered at 1.0 mg/kg body weight once every 2 weeks as an intravenous infusión) (Figure 1).

In 2016, migalastat was approved by the FDA and is available to treat FD patients with “amenable” mutations [12,15] (Figure 1).

Figure 2 shows a schematic migalastat mechanism of action (2A). In renal tissue, without specific treatment for FD, there is a pathological lysosomal Gb3 accumulation and impaired renal function (2B) while, when α-GalA enzymatic activity is restored, the accumulation of Gb3 and derivative metabolites is prevented, and this allows adequate renal function (2C). 

A biomarker is a defined characteristic that is measured as an indicator of normal biological processes, pathogenic processes or responses to an exposure or intervention. A biomarker may be used to see how well the body responds to a treatment for a disease or condition. [16]. As i) Fabry disease is a rare disorder that frequently causes kidney damage and ii) a new therapeutic is currently available, it is necessary to review which biomarkers are useful for nephropathy follow-up among Fabry “amenable” patients receiving migalastat [17].

The aim of this work is to carry out a bibliographic search regarding which biomarkers are useful for monitoring FD patients and nephropathy treated with migalastat.

## 2. Materials and Methods

- Literature review

The literature search was conducted in the following electronic databases: MEDLINE, EMBASE, SCOPUS, Cochrane, and Google academic. The search terms used were “Fabry disease”,“renal disease”, “biomarker”, and “migalastat”

- Inclusion criteria

Prospective studies in which renal biomarkers were the dependent variable or criterion were included. Studies with at least 6 months of follow-up were included (because it is the recommended average period of evaluation for FD kidney damage) [3].

- Exclusion criteria

Publications that were “preliminary reports” or “sub-analyses” of other works were excluded to avoid duplication of reported cases.

- Procedure

Two investigators conducted the search independently, and a third investigator corroborated the information overlap. Finally, we recorded relevant information in an ad hoc database to sort the publications and summarize the main results.

## 3. Results

Four studies met the inclusion criteria [18,19,20,21]; in addition, two studies were analyzed [22,23] that are preliminary reports of other publications, but they are the only published studies that include the biomarker “renal histology” during the follow-up of FD patients treated with migalastat [Table 1].

MEDLINE = Total *n*= 11, post analysis *n* = 2 [18,19]; 3 studies were excluded because they were preliminary reports of other studies [22,24,25]; 4 studies were excluded because they were sub-analysis of other studies [23,26,27,28] and 2 studies were excluded because they did not meet the inclusion criteria [29,30]. EMBASE = Total *n* = 0; SCOPUS = Total *n* = 0, Cochrane Total *n* = 0 and Google Academic = Total *n* = 191, post analysis *n* = 2 [20,21]; (11 studies were overlapped with MEDLINE, both included and excluded; 1 study was excluded because it was a preliminary report of other study [25], 1 study was excluded because it did not have renal biomarkers among its dependent variables [31]; 1 study was excluded because its sub-analysis of other studies [32]; 173 studies were excluded for not meeting inclusion criteria).

Data discrepancies between both researchers: 1 study was retrieved by one of the researchers and not by the other [33]. 

### 3.1. Serum Creatinine and Glomerular Filtration Rate

Feldt-Rasmussen et al. reported the results of ATTRACT (Study AT1001-012; NCT01218659) at 30-month to follow-up; 48 patients aged 16 to 74 years, enrolled from 25 centers in 10 countries, classified into two groups: Group 1 (Group Migalastat) (*n* = 33): patients who received migalastat during the 30-month and, Group 2 (Group ERT-Migalastat) (*n* = 15): patients who received ERT during the randomized period (18-month) discontinued ERT and started treatment with migalastat (12-month). In this study, measures of the annualized rate of change in eGFR were assessed using the Chronic Kidney Disease Epidemiology Collaboration equation (eGFRCKD-EPI) at baseline, months 1, 3, 6, 9, 12, 15, 18, 19, 21, 24, and 30 were informed [18]. 

A study of Supplementary Material shows participant’s individual information, classified by group; it is observed that adult patients with pathogenic GLA gene mutations that cause both “late-onset” phenotype as well as the “classic” FD phenotype were included [18]. 

Authors reported stabilization of renal function determined by the annualized rate of change in eGFR CKD-EPI in both groups of patients. In Group 1 (Migalastat), the mean annualized rate of change from baseline to month 30 was −1.7 mL/min/1.73 m^2^ (for males: −2.1 mL/min/1.73 m^2^ and for females: −1.4 mL/min/1.73 m^2^). In Group 2 (ERT-Migalastat): −2.1 mL/min/1.73 m^2^ (for males: −5.7 mL/min/1.73 m^2^ and for females: −0.3 mL/min/1.73 m^2^) [18]. 

Müntze et al. described in 14 patients follow-up during 12-months of treatment with migalastat sub-classified into two groups: (1) the treatment-naive group (received 12 months of migalastat) and (2) the group that had previously received ERT and then switched to migalastat [21].

In total population the mean eGFR CKD-EPI was 87 (75.5–102) mL/min/1.73 m^2^ at baseline and 78 mL/min/1.73 m^2^ at 12-months; and 76 to 72 (73.5–95.5) mL/min/1.73 m^2^ in Naive group. The authors performed a sub-analysis of these results and reported that of the total population: (i) women, (ii) the group switch, and (iii) patients with the N215S mutation presented stable eGFR during follow-up (most FD patients with the genetic variant N215S-mostly late-onset cardiac form do not present renal functional changes, as it is also the case of benign variants); but that a decreased GFR was present in the rest of patients, although within this last group of patients, patients with mutations likely benign such as N139S were included, which does not cause kidney damage. In these cases, an appropriate genotype–phenotype correlation analysis is necessary so as not to attribute treatment failure to patients who are not going to develop kidney damage due to FD, such as patients with late-onset cardiac phenotypes or benign GLA gene variants carriers [21].

Lenders et al. [19] reported results of 54 FD adult patients (mean age: 45 years) treated with migalastat during 24 months of follow-up. Similarly to the previously described studies, patients who started migalastat for FD treatment (Naive Group) and patients who had previously received ERT (Switch Group) were included. Among the included population, 31 patients suffered from arterial hypertension at baseline, 37 patients were on anti-hypertensive medication, and 7 patients had diabetes mellitus. The study’s primary purpose was to evaluate the left ventricular mass (LVMi) and, among the secondary endpoints, included the GFR during the follow-up period [19]. This trial showed that patients with additional (cardiovascular) risk factors suffered from a more prominent eGFR decline after 1 year of migalastat treatment, an expected result due to the high prevalence of cardiovascular risk factors in the included population. Logistic regression analysis of variables associated with a greater decrease in GFR showed that: patients receiving angiotensin-converting enzyme (ACE), AT1, or aldosterone inhibitors showed a significant loss of eGFR after 12 months (−6.1 ± 7.2 mL/min/1.73 m^2^; *p* < 0.0001) and 24 months (−9.3 ± 13.4 mL/min/1.73 m^2^; *p* = 0.0003), whereas patients receiving other hypertensive drugs were stable after 12 months (−5.4 ± 8.6 mL/ min/1.73 m^2^; *p* = 0.2322) and 24 months (−1.3 ± 7.7 mL/min/1.73 m^2^; *p* = 0.7025) [19].

In this study, patients with GLA genetic variants causing the “late-onset” FD phenotype (R118C, S126G, M296V) or GLA VUS (variants of unknown significance) (T385A) were included. Because these mutations are not the cause of kidney damage in affected patients at the mean age of the included population (45 years) it is appropriate to analyze causes of nephropathy other than FD influencing the worse renal prognosis and also analyze the population sub-classified in FD phenotypes [19].

Riccio et al. reported follow-up results of 7 men (5 classic, 2 late-onset) in whom a switch from ERT to migalastat was indicated [20]. This study reported data from 12 months of follow-up under treatment with migalastat. GFR remained unchanged during the follow-up period, both in patients who had normal or decreased renal function at baseline. [20]

### 3.2. Albuminuria/Proteinuria

The ATTRACT study design defined renal adverse events such as elevated protein ≥ 300 mg. Authors reported that no significant change from baseline in 24 h urine protein was observed in Group 1 (Migalastat) from 0 to 30 months, and in Group 2 (ERT-Migalastat) during both the initial 18-month period (ERT) and the subsequent 12-month open-label extension period (migalastat). [18]

In the FAMOUS study, no changes in albuminuria were detected after 12 or 24 months of treatment with migalastat in either females or males [19]. This study described the results of 54 FD adult patients who started migalastat for FD treatment (Naive Group) and patients who had previously received ERT (Switch Group) were included [19].

The AFFINITY group reported that median proteinuria showed a significant decrease during 12-months of follow-up (*p* = 0.048) in affected males [20]. This study described the results of 7 men who switched from ERT to migalastat prior to 12-month follow-up [20].

### 3.3. Plasma Lyso-Gb3

The ATTRACT Study reported decreased Lyso-Gb3 plasma levels in Group 1 (Migalastat) and Group 2: (ERT-Migalastat), but the study design did not include an analysis of the correlation of these low Lyso-Gb3 levels and kidney function parameters [18].

Müntze et al.: Lyso-Gb3 values showed a decreasing trend; median values declined from 10.4 (IQR 6.5–13.8) ng/mL to 6.1 (IQR 4.0–14.7) ng/mL (*p* = 0.319) in the total patient cohort at the 1-year follow-up (*n* = 14). A significant reduction was observed in the therapy-naive subgroup, from 10.9 (IQR 7.0–15.7) ng/mL to 6.0 (IQR 4.0–8.1) ng/mL (*p* = 0.021, *n* = 8) [21].

Lenders et al. reported that plasma lyso-Gb3 levels remained stable in females and males during the 24-months to follow-up [19].

Plasma lyso-Gb3 levels showed a significant reduction during the follow-up in the cohort studied by Riccio et al. [20]. 

The studies of Feldt-Rasmussen, et al., Müntze et al., and Riccio et al. reported that plasma lyso-Gb3 decreases during follow-up in patients treated with migalastat, while Lenders et al. showed plasma Lyso-Gb3 stabilization. In these four studies, a correlation between plasma Lyso-Gb3 and renal function biomarkers was not studied.

### 3.4. α-GalA Activity

In male patients included in ATTRACT Study [18], α-GalA activity was measured. Male patients in Group 1 (Migalastat) had a significant increase in α-GalA activity from baseline to month 30 (mean change from baseline: 4.1 nmol/h/mg; median: 3.5; 95% CI: 1.9, 6.3). In males in Group 2 (ERT-Migalastat) α-GalA activity remained stable (mean: 3.4 nmol/h/mg; median: 2.3; 95% CI: −2.7, 9.4). The study design did not perform a correlation analysis between α-GalA activity and renal function parameters.

Müntze et al. reported an α-GalA activity significantly increased from a median of 0.06 to 0.2 nmol/min/mg protein (*p* = 0.001) in 14 patients during 1-year follow-up, receiving migalastat, either with or without prior ERT [21]. Similar to ATTRACT, the study design did not perform a correlation analysis between α-GalA activity and renal function parameters.

In the FAMOUS study, enzymatic α-GalA activities were constant in females and males at Months 12 and 24 compared to baseline [19]. In this study, the correlation between α-GalA enzymatic activity and renal function biomarkers was not studied either.

### 3.5. Renal Histology

Two studies show results regarding renal histology in patients receiving migalastat treatment; both are preliminary reports of another primary study of our systematic search [22,23].

Germain et al. reported that Gb3 deposits changes in kidney interstitial capillaries (KIC). Their results showed that in the 6-months primary end-point analysis, 13 of 32 patients (41%) who received migalastat had a response (≥50% reduction in number of Gb3 inclusions per KIC) and the median change in KIC Gb3 from baseline was −40.8% with migalastat compared versus −5.6% with placebo. The reduction in KIC Gb3 at 6 months remained stable after an additional 6 months of treatment. In this trial, authors reported results of 23 kidney biopsy samples after 12 months of migalastat treatment: patients showed decreases in Gb3 in glomerular podocytes (5 of 23 samples, 22%), endothelial cells (6 of 23 samples, 26%), and mesangial cells (11 of 23 samples, 48%). None of the samples had increases [22].

Mauer et al. reported the results of 8 male patients with “classic” FD with a mean age of 42 years, receiving treatment with migalastat, with a renal biopsy sample prior to initiation and at 6 months of follow-up. The authors show a beneficial effect on the podocyte cell, with decreased cytoplasmic inclusions of Gb3, decreased podocyte volume and a tendency to correlate these findings with urinary protein excretion without statistical significance. Of note, the authors reported a significant correlation between decreased plasma Lyso-Gb3 and a decline in podocyte volume and decreased cytoplasmic podocytes Gb3 inclusions clearance. The limitation of the study is the short follow-up period, which does not allow definitive conclusions regarding this topic [23].

## 4. Discussion

FD patients treated with phamacological chaperones present stable eGFR, while there are no patients reported who has presented ESRD during follow-up. In this regard, patients with “amenable” mutations and kidney damage due to FD, have a favorable response to migalastat and eGFR is correlated with this favorable evolution, but patients must be adequately characterized prior to starting treatment since, as has been discussed, misinterpretation of response to migalastat may occur when: (i) cardiovascular risk factors or other kidney damage causes are not adequately treated or considered, (ii) patients are receiving drugs that can modify eGFR, (iii) patients who carry GLA mutations or variants that typically do NOT affect kidney function. In these cases, it is not appropriate to attribute kidney damage to FD or poor renal response to migalastat. Renal biopsy should be performed in these selected patients, to clarify the etiology of poor renal evolution.

Proteinuria has not been included in trial designs of all currently available studies; although it has remained stable on studies in which it was included; no study found a correlation between proteinuria and renal prognosis. The presence of severe proteinuria (>1 g/24 h) has been for a long time, a risk factor recognized for CKD progression in FD. In patients treated with migalastat, proteinuria/albuminuria is a biomarker of nephropathy added to FD and is a criterion for performing a renal biopsy to diagnose another kidney disease cause, different to FD. 

Increased α-GalA enzyme activity and decreased Lyso-Gb3 have been described in patients of both genders, both “classic” and “late-onset”; but no study found a correlation between renal evolution and those biomarkers among “amenables” FD patients. The work of Mauer et al. described a relationship between Lyso-Gb3 and Gb3 deposits in podocytes and consequently with proteinuria. However, to date, there is no evidence to support the usefulness of these biomarkers in monitoring the therapeutic response to migalastat in FD nephropathy.

Regarding renal biopsy, although it has been reported that there is an improvement in renal histological damage due to migalastat treatment in patients affected by “amenable” mutations, including men with the classic disease phenotype, the routine use of renal biopsy is not adequate, since there are non-invasive biomarkers that correlate with kidney damage evolution, leaving its indication for selected cases, as discussed above.

Other biomarkers, for example, podocyturia, micro-RNAs, and urinary mRNA have been described in FD nephropathy, although they are not available in routine practice [6,34,35]. There is no current evidence regarding its use in FD patients receiving migalastat follow-up. Its use would be relegated to basic research, helping to increase the underlying mechanisms of FD renal damage understanding.

## 5. Conclusions

To date, eGFR is the main and only useful biomarker, that has been shown in clinical studies for nephropathy monitoring among Fabry “amenable” patients receiving migalastat. However, albuminuria/proteinuria could be useful to assess the indication for concomitant medication or renal biopsy in selected patients.

## Figures and Tables

**Figure 1 genes-13-01751-f001:**
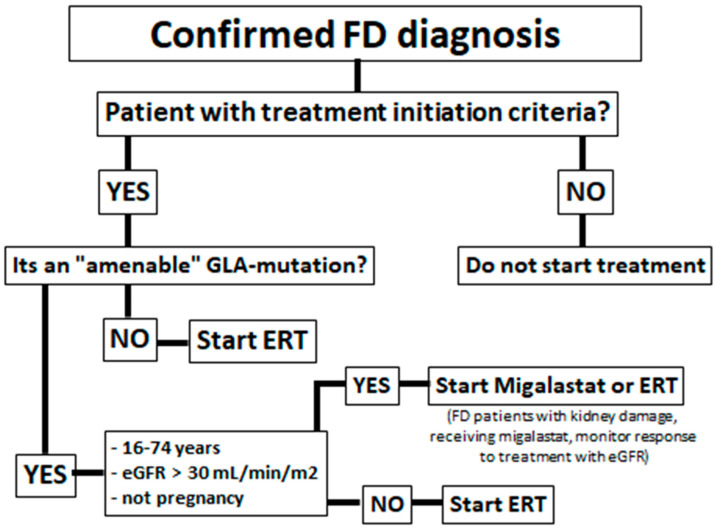
Algorithm for the indication of migalastat in Fabry disease patients.

**Figure 2 genes-13-01751-f002:**
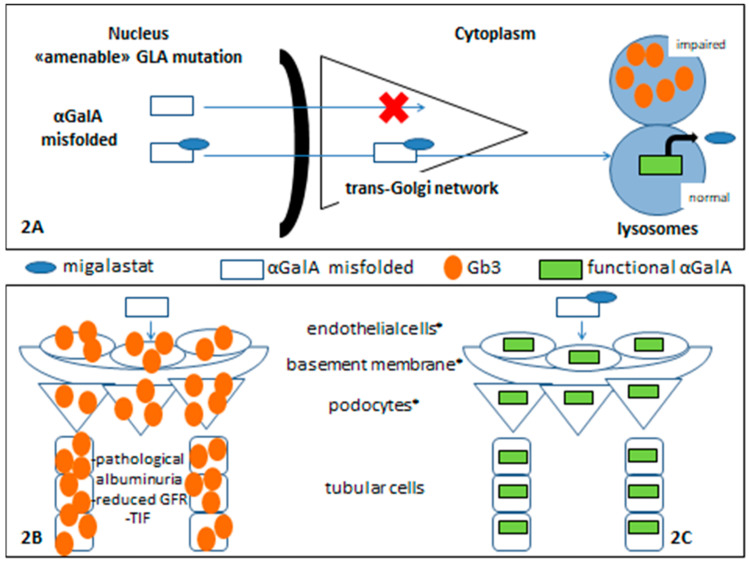
Migalastat in Fabry nephropathy. **References:** * glomerular filtration barrier components.

**Table 1 genes-13-01751-t001:** Studies analyzed.

Author (year)	Uni/ Multi-Center	Follow-Up	N	Included Both Genders	Included Classic and Late-Onset	Renal Biomarker	Other Biomarkers
Feldt-Rasmussen, et al. (2020) [18]	Multicenter (10 countries) *	30-months	48	Yes	Yes	eGFR proteinuria	-cardiac function -stroke/TIA -α-GalA -Lyso-Gb3
Müntze et al. (2019) [21]	Unicenter (Germany)	12-months	14	Yes	Yes	eGFR proteinuria	-cardiac function - α-GalA -Lyso-Gb3
Lenders et al. (2022) [19]	Multicenter (Germany)	24-months	54	Yes	Yes	eGFR proteinuria	-LVMi -stroke/TIA -MSSI -DS3 -Lyso-Gb3
Mauer et al. (2017) [23]	Unicenter (United States)	6-months	8	Not (only males)	Not (only classic)	eGFR proteinuria renal histology	-Lyso-Gb3
Germain et al. (2016) [22]	Multicenter (11 countries) **	12-months	50	Yes	Yes	eGFR proteinuria renal histology	-cardiac function - α-GalA -Lyso-Gb3 -GSRS -SF-36 -PS-SF
Riccio et al. (2020) [20]	Unicenter (Italy)	12-months	7	Not (only males)	Yes	eGFR proteinuria	-cardiac function -neurologic changes -MSSI -SF-36

Abbreviations: eGFR: estimated glomerular filtration rate; α-GalA: α-galactosidase A; LVMi: left ventricular mass index; TIA: transient ischaemic attack; MSSI: Mainz Severity Score Index; DS3: Disease Severity Scoring System; GSRS: Gastrointestinal Symptom Rating Scale; SF-36: the Medical Outcomes Study 36-Item Short-Form Health Survey version 2; PS-SF: pain-severity component of the Brief Pain Inventory–Short Form. References: * Australia, Austria, Belgium, Brazil, Denmark, France, Italy, Japan, the United Kingdom, and the United States; ** France, the United Kingdom, Australia, Canada, Brazil, the United States, Italy, Turkey, Argentina, Denmark, Egypt, Spain.

## Data Availability

Not applicable.

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
