# Peer review of "Biomarkers for Monitoring Renal Damage Due to Fabry Disease in Patients Treated with Migalastat: A Review for Nephrologists"

_genes, 2022, doi:10.3390/genes13101751_

Round 1
Reviewer 1 Report
Manuscript ID: genes-1892914
Type of manuscript: Review
Title: Biomarkers for monitoring renal damage due to Fabry disease in patients treated with migalastat. Systematic review for nephrologists
The objective of the manuscript is to review useful biomarkers for the follow-up of patients with "amenable mutation" and nephropathy caused by FD. The authors performed a systematic literature search on published articles useful for the primary objective of the work, according to their inclusion criteria. They found 6 publications that were analyzed and discussed. The authors summarize the results of these selected publications focusing on the relationship of migalastat treatment and renal biomarkers. Their conclusion is that eGFR is the only useful biomarker for FD nephropathy monitoring, at least to evaluate migalastat response.
Line 70: pharmacological chaperone instead of chemical
Line 71: migalastat was already approved in EU by EMA in May 2016: https://www.ema.europa.eu/en/medicines/human/EPAR/galafold
Line 74: please rephrase the sentence, but not just quoting the biomarker definition
Line 91: included
Line 97: please, clearly explain the publications that finally meet the criteria and refer to the Table 1
Line 111: delete Figures, Tables and Schemes
In Table 1, I would not include the article of Di Stefano et al [34] because with the initial eGFR and proteinuria you do not have the diagnosis of FD (no FD specific lesions were described in the renal biopsy) and after the diagnosis the patient already had received a kidney transplant.
Line 113: Delete references and change to “abbreviation”. In order to maintain uniformity, refer to eGFR always along the paper and not sometimes just GFR.
Line 131: Müntze et al describe …
Lines 134-135: Delete this sentence. Isolated serum creatinine is not a valid biomarker of renal function.
Line 139: most FD patients with the genetic variant N215S (mostly late-onset cardiac form) do not present renal functional changes, as it is also the case of benign variants.
Line 143: if those patients do not develop kidney damage, we can’t in anyway attribute that to a treatment failure.
Line 157: when comparing eGFR loss in patients receiving ACE, AT1, and aldosterone inhibitors, with those patients receiving other anti-hypertensive drugs (please, describe what type of drugs), it is very important to know the degree of proteinuria in both cohorts. The first group could be receiving anti-proteinuric drugs because of severe proteinuria, which is a risk factor for eGFR progressive decline.
Line 161-162: patients with GLA genetic variants causing the "late-onset" FD phenotype… or GLA VUS (variants of unknown significance) … were included.
Lines 170-171: delete the sentence. This patient was receiving migalastat after ERT but had already received a kidney transplant and thus renal biomarkers are not valid for this study.
Line 173: rephrase “with the elevated protein >300 mg”
Line 178: describe FAMOUS
Line 180: describe AFFINITY. “significant” decrease
Line 192: significant reduction during the follow-up
Line 213: Two studies show results…
Lines 233-234: What is the meaning of this sentence? GFR estimated by equations that include creatinine (eGFR) has been shown correlation with Fabry nephropathy evolution among “amenable” patients. eGFR is just a measure of the glomerular function in any type of nephropathy. Please, modify your expression about patients with FD with amenable mutations.
Line 235: What is the meaning of this sentence: While there are no patients reported in the literature who have presented ESRD during follow-up, eGFR correlates with nephropathy evolution … (Please, explain the significance of correlation with FD nephropathy evolution)
Line 237: kidney damage
Line 240: misinterpretation
Line 241: receiving
Line 243: What is the proof that the nephropathy is not secondary to FD? Or adverse effect of migalastat?
Line 247: the presence of severe proteinuria (> 1 g/24 h) has been for a long time recognized as a risk factor for CKD progression in Fabry disease.
Line 248: criterion
Line 251: both "classic" and "late-onset" FD
Line 252: even if no correlation has been found in the described studies among Lyso-Gb3 and renal function, the work of Mauer et al show a relationship between Lyso-Gb3 and Gb3 deposits in podocytes and consequently with proteinuria. Thus, more data about Lyso-Gb3 and renal function biomarkers could eventually bring some light to the pathophysiological consequences of Gb3 deposits and exposure to Lyso-Gb3.
Line 253: response
Line 256: routine
Line 257: adequate
Line 258: leaving its indication for selected cases….
Line 259: see the previous comment about plasma Lyso-Gb3 and podocyte deposits
Line 265: eGFR is not good enough as an early nephropathy biomarker. The periodic use of albuminuria/proteinuria may help to predict response to ERT or migalastat, and indicate the use of concomitant treatments, to delay the progression of FD nephropathy.
References: please look up the instructions for the authors (Abbreviated Journal Name Year, Volume, page range). References #22 and 31 are repeated.
Author Response
All of Reviewer 1's suggestions have been included in the current version of the manuscript. In the attached (word) file we respond 1 to 1 the suggestions. Highlighted in yellow

Reviewer 2 Report
In this concise review, the authors reviewed representative studies following up the renal biomarkers in Fabry Disease (FD) patients treated with magalastat. The authors describe the common FD renal biomarkers in detail and provided specific and concise example from the representative studies. This compilation of literature thus provides useful insights for both the scientific community and the practitioners.
However, the authors should consider addressing the following issues in this manuscript:
Major points:
1. The authors should consider giving clear and convincing explanations why:
a) the inclusion threshold is set to 6 month follow-up period,
b) the preliminary reports of other studies are excluded,
c) the sub-analysis of other studies are also excluded from this literature review;
as all these published reports may contain important information regarding these renal biomarkers and thus could give a more comprehensive view of this field and a better comparison.
2. It would be ideal if the authors could, in the introduction, elaborate the summarization of current clinical management of FD including magalastat, as well as provide a figure depicting briefly the molecular mechanisms of FD in the kidney, the action mode of magalastat, and the connections to the common biomarkers.
3.Table 1: the authors may consider including more information from the selected studies to benefit the scientific community, e.g., beside renal biomarker, what other biomarkers are studied in this study; which organs are also being analyzed; age; country; year, etc.
4. The authors may consider proposing new avenues/biomarkers for future research in this field in the discussion section.
Minor points:
1. The authors should consider introducing the basic concept of biomarkers in own words, line 74-75.
2. The authors should refine the description of the method part, e.g., line 92-95 and the subsequent results may not be necessary.
To summarize, I agree that this paper may be relevant for publication in Genes. The manuscript covers representative recent discoveries in this field and could be improved by addressing the above issues, I therefore recommend reconsideration of the manuscript after major revision.
Author Response
All the suggestions of the #2 reviewer were included in the manuscript (highlighted in light blue)

Round 2
Reviewer 2 Report
In this version, the authors addressed several issues mentioned in the previous comments, including:
Major points:
1. briefly listed the main reasons why a) the inclusion threshold is set to 6 month follow-up period, & c) the sub-analysis of other studies are also excluded from this literature review;
3. updated Table 1 with other biomarkers are studied in this study country; year, etc. (Yet, Gemain et al., 2016, country information is missing)
4. extended a bit on the discussion of new avenues/biomarkers for future research in this field.
Minor points:
1. introduced the basic concept of biomarkers in own words.
2. reorganized the description of the method part.
However, the authors did not give reasonable response to the following points:
Major point:
2. in the introduction, elaborate the summarization of current clinical management of FD including magalastat, as well as provide a figure depicting briefly the molecular mechanisms of FD in the kidney, the action mode of magalastat, and the connections to the common biomarkers.
To summarize, the authors have made several modifications to the previous version, but the manuscript fails to address the remaining issues stated above. I therefore recommend reconsideration of the manuscript after major revision.
Author Response
Respnse:
3. updated Table 1 with other biomarkers are studied in this study country; year, etc. (Yet, Gemain et al., 2016, country information is missing). Table 1 was modified with the author's suggestion (highlighted in green)
Major point:
- in the introduction, elaborate the summarization of current clinical management of FD including magalastat, as well as provide a figure depicting briefly the molecular mechanisms of FD in the kidney, the action mode of magalastat, and the connections to the common biomarkers. Added figure 2 (scheme showing reviewer's suggestion,highlighted in green)
